# Local Vibrational Therapy for Essential Tremor Reduction: A Clinical Study

**DOI:** 10.3390/medicina56100552

**Published:** 2020-10-21

**Authors:** Silvijus Abramavičius, Mantas Venslauskas, Antanas Vaitkus, Vaidotas Gudžiūnas, Ovidijus Laucius, Edgaras Stankevičius

**Affiliations:** 1Laboratory of Preclinical Drug Investigation Institute of Cardiology, Lithuanian University of Health Sciences 1, 44307 Kaunas, Lithuania; v.gudziunas@gmail.com (V.G.); ovidijus.laucius@gmail.com (O.L.); edgaras.stankevicius@lsmuni.lt (E.S.); 2Institute of Mechatronics, Kaunas University of Technology, 44249 Kaunas, Lithuania; mantas.venslauskas@ktu.lt; 3Department of Neurology, Lithuanian University of Health Sciences, 44307 Kaunas, Lithuania; antanas.vaitkus@kaunoklinikos.lt

**Keywords:** essential tremor, local hand-arm vibration, mechanical vibration, vibrational therapy, essential tremor treatment

## Abstract

*Background and objectives:* tremor is an unintentional and rhythmic movement of any part of the body that is a typical symptom of Essential Tremor (ET). ET impairs the quality of life of patients and is treated with pharmacotherapy. We investigated the tremor reduction efficacy of an innovative vibrational medical device (IMD) in ET patients. *Materials and Methods*: we conducted a prospective, single-center, single-arm, pragmatic study in ET patients with an extended safety study to evaluate the efficacy and safety of the Vilim Ball—a local hand-arm vibration device that produces vibrations in the frequency range of 8–18 Hz and amplitude from 0 to 2 mm. The primary endpoint was the decrease in the power spectrum after device use. The secondary endpoints were safety outcomes. *Results:* In total, 17 patients with ET were included in the main study, and no patients withdrew from the main study. The tremor power spectrum (m^2^/s^3^ Hz) was lower after the device use, represented as the mean (standard deviation): 0.106 (0.221); median (Md) 0.009 with the interquartile range; IQR, 0.087 vs. 0.042 (0.078); Md = 0.009 with the IQR 0.012; Wilcoxon signed-rank test V = 123; and *p* = 0.027. Seven patients reported that vibrational therapy was not effective. Two patients reported an increase in tremor after using the device. In the extended safety study, we included 51 patients: 31 patients with ET and 20 with Parkinsonian tremor, where 48 patients reported an improvement in tremor symptoms and 49 in function. No serious adverse events were reported, while two patients in the Parkinsonian tremor group reported a lack of efficacy of the proposed medical device. *Conclusions:* the device reduces essential tremor in some patients and is safe to use in ET.

## 1. Introduction

Tremor is an unintentional and rhythmic movement of any part of the body that is a typical symptom of Essential Tremor (ET) [1]. The first-line treatment of ET usually consists of pharmacotherapy: Propranolol and Primidone (First-line treatments); Topiramate and Gabapentin (Second-line treatments); Clonazepam and Mirtazapine (Third-line treatments). To date, the beta-blocker propranolol and the antiepileptic drug primidone are the drugs with the highest efficacy in the treatment of ET [2]. However, the benefits of pharmacotherapy are limited. Additional therapies are surgical treatment, such as Thalamotomy and Deep brain stimulation [3]. The crude prevalence rates of essential tremor in adults range from 0.4% to 6% [3,4,5,6]. ET affects approximately 1% of the population and 4–5% of people aged over 65 years [5]. ET seems to have a bimodal age of onset [6]. A younger group of patients develops ET before they reach 24 years of age, while older ET patients develop ET after they reach 46 years of age [7]. We developed a device for the treatment of essential tremor: the Vilim Ball is a therapeutic device which was created to reduce hand tremor in patients with ET. 

Various novel therapeutic moieties are currently under development for the ET. L-octanol, a long-chain alcohol, was shown to reduce the tremor [8,9,10]. Perampanel, a selective, non-competitive α-amino-3-hydroxy-5-methyl-4-isoxazolepropionic acid (AMPA) receptor antagonist that blocks glutamate activity in postsynaptic AMPA receptors was shown to improve the tremor symptoms too [10].

Medical device development for the ET patients is afoot [11]. Non-invasive neuromodulation through ulnar and radial nerve stimulation was shown to be effective and had a favorable safety profile in the ET patients in a sham-controlled clinical trial [12]. Transcranial magnetic stimulation is also being explored as a treatment option in the aforementioned patient population [13].

We sought to evaluate the efficacy and safety ratio of our device for the ET treatment.

## 2. Materials and Methods

### 2.1. Design of the Clinical Study

We conducted a prospective, single-center, pragmatic clinical study [14], similar to routine medical care. In this study, the patients acted as their own controls. We assumed that a single-arm design (without sham control) is appropriate because the primary endpoint is objective, and it is unlikely that patients with essential tremor could achieve spontaneous remission [15]. We recorded the tremor data for 1 min before the intervention and 1 min after the intervention with an accelerometer and transformed the data with Fourier-transformation for power spectrum analysis [16]. The change in power spectrum analysis was the primary efficacy endpoint.

### 2.2. Data and Statistical Analysis

We presented data as the mean (standard deviation) or as the median and interquartile range if the data were asymmetrically distributed. We assessed the data symmetricity with the Normal Q-Q plots. We chose the significance level of *p* < 0.05. We compared the differences between the repeated measures of the primary endpoint with the Wilcoxon signed-rank test. We performed the statistical analysis with R version 3.5.0.

All analyzed patients were included in the Efficacy and Safety analysis set. The efficacy and safety analysis sets were equivalent to the Intent-to-Treat (ITT) Analysis Set. 

### 2.3. Ethical Statement

The clinical investigation plan was developed in line with the EN ISO 14155-1:2009 Clinical Investigation of Medical Devices for Human Subjects—General Requirements, which details the general requirements for the conduct of clinical investigations and EN ISO 14155-2:2009 Clinical Investigation of Medical Devices for Human Subjects. Research with human subjects was performed in line with the Helsinki Declaration adopted by the 18th World Medical Assembly in Helsinki, Finland, in 1964, as last amended by the World Medical Assembly. All measures relating to the protection of human subjects were taken with the core principles of the Helsinki Declaration in mind. A general principle was held, “the rights, safety, and wellbeing of clinical investigation subjects shall be protected consistent with the ethical principles laid down in the Declaration of Helsinki” (EN ISO 14155-1:2009). Bioethics permission (BE-2-90) from the regional research ethics committee was received.

### 2.4. Investigated Medical Device (IMD)

Vilim Ball is a therapeutic device that was created to relieve ET. The Vilim ball device emits low-frequency mechanical vibrations and generates mechanical vibration in the frequency range of 8–18 Hz (amplitude: 0–2 mm). The Vilim Ball device is a non-invasive portable physiotherapeutic device. The device is intended to be used at home or in a hospital environment (Figure 1). 

### 2.5. Efficacy and Safety Evaluations

#### 2.5.1. Efficacy Evaluation

The subjects were asked to hold a smartphone in the dominant hand before vibrational therapy and after it. Tremor data were captured by using a mobile application that was designed to collect raw data (signal frequency 100 Hz) provided by the accelerometer. The primary endpoint was the decrease in the power spectrum (m^2^/s^3^ Hz) [17,18] after use of the IMD. The baseline tremor power spectrum was evaluated. After that, vibrational therapy was performed for 5 min for each patient, and the postinterventional tremor power spectrum was evaluated. The difference between baseline and postinterventional tremor power spectrum was determined (as defined in the statistical analysis paragraph). Collected data were filtered to evaluate the range of 4–12 Hz, which is hand tremor frequency with diagnosed essential tremor.

#### 2.5.2. The Secondary Efficacy Endpoint

The secondary efficacy endpoint was the patient’s assessment of the vibrational therapy effectiveness in a Likert scale. Additionally, the patient’s assessment of the tremor intensity before the intervention was assessed (Figure 2).

#### 2.5.3. Exploratory Endpoint

A TETRAS questionnaire was used to evaluate the tremor intensity before intervention [19] to establish a relationship between the accelerometry data and the TETRAS estimate.

#### 2.5.4. Sample Size

The sample size was estimated during the conduct of the pilot part of the study after the evaluation of 5 patients before and after the use of the IMD. This study was designed to have an α error probability = 0.05; power (1-β error probability) = 0.8; the expected size of difference was at least 0.01; and the effect size was at least 1. The total estimated sample size was 17 patients. The sample size was assessed with G*Power 3.1 [20].

### 2.6. Selection of Study Population 

The study population included subjects diagnosed with Essential tremor, who were randomly selected at The Hospital of Lithuanian University of Health Sciences (LSMU) Kauno klinikos, Neurology Department and Outpatient Division of Nervous System Diseases (Table 1).

### 2.7. Additional Extended Safety Study

A retrospective cross-sectional study was planned during the clinical development of the IMD. The study population consisted of adult subjects with neurological diseases that cause movement disorders of hands, who signed informed patient consent during the meeting with the investigator. The primary efficacy outcome was the Patient-Reported Outcome based on a non-validated patient telephone questionnaire [21]. Safety assessments were descriptive and included a summary of adverse events (AEs) if they were observed. The safety assessment was conducted by the attending physician. The treating physician determined the need for physical examinations, vital signs, electrocardiography (ECGs), and laboratory measurements (haematology, chemistry, and urinalysis) to assess safety. The Medical Dictionary for Regulatory Activities (MedDRA) nomenclature were used to code AEs.

## 3. Results

### 3.1. Clinical Efficacy Assessment

In total, 6 men and 11 Caucasian women with essential tremor were included in this study from 1 May 2017 and 31 December 2017. The mean age was 70.82 (8.03) years. Seven people reported the use of drugs for ET. Five people reported alcohol consumption. Seven people had family members that had essential tremor. The primary endpoint was the tremor power spectrum. The distributions of the primary endpoint before the intervention (PE1) and after the intervention (PE2) were highly asymmetric: PE1 mean 0.106 ± 0.221 median (Md) 0.009 (interquartile range, IQR 0.087) and PE2 mean = 0.042 ± 0.078; Md = 0.009 (IQR 0.012); Wilcoxon signed-rank test V = 123; *p* = 0.027 (Table 2). Four cases were identified as IMD inefficacy. Seven patients reported that vibrational therapy was not effective. Two patients reported an increase in tremor after using the device. No other adverse events were reported by the patients included in this study. No serious adverse events were identified by the study investigators.

### 3.2. Exploratory Analysis in the Clinical Efficacy Assessment

The primary endpoint tremor power spectrum (PE1) before the intervention was correlated with the TETRAS score. Spearman’s rank correlation: rho 0.765, *p* < 0.001. 

### 3.3. Additional Extended Safety Study

A cross-sectional study was performed in patients with Essential and Parkinsonian tremor that used the Vilim Ball prototype to assess the safety and efficacy of the Vilim Ball. The investigation was held by scientists of interested party MB “Fidens” company, which invented the Vilim Ball. Bioethics permission (BE-2-90) from the regional research ethics committees was received. In total, 51 patients with the mean (standard deviation) age of 66.9 (16.28) were included (31 in the essential tremor and 20 Parkinsonian tremor) in the study between 1 September 2018 and 31 September 2019. The primary efficacy outcome was the Patient-Reported Outcome based on a non-validated patient telephone questionnaire [21]. The secondary outcome was the occurrence of adverse events. Forty-eight patients reported improvement in tremor symptoms, and 49 reported an improvement in function. The patients used the Vilim Ball for 7.63 (5.41) months. Thirty-eight patients were able to report the duration of improved function, which was 90.79 (68.83) minutes. Two patients reported a lack of efficacy of the proposed medical device during the study. No other serious adverse events were reported.

## 4. Discussion

In our study, we found that local vibrational therapy is an effective treatment option in some patients. Our device has novel technological characteristics. However, previous attempts were made to develop medical devices for the treatment of ET. Theoretical (conceptual) articles have been published by several authors regarding the use of the self-balancing device [22] and exoskeleton power-assist robot to treat essential tremor [23]. However, these concepts were not clinically developed. Other devices went beyond the conceptual phase, and their efficacy was assessed in clinical trials. A pilot randomized sham-controlled pilot trial was conducted to see if local vibration is effective in patients with essential tremor. The pilot study revealed promising results, as the motor performance, evaluated with the Archimedes spiral drawing task, improved after stimulation, when compared against baseline [12]. This research (with the same device) was followed with a pivotal study, where it was shown that subjects who received peripheral nerve stimulation did not show a significantly larger improvement in the Archimedes spiral task compared to sham but did show a significantly greater improvement in upper limb TETRAS tremor scores (*p* = 0.017) compared to sham. No significant adverse events were reported; 3% of subjects experienced mild adverse events [24]. Vibrating Gaussian noise emitting manipulandum has been shown to improve motor performance. A pilot study showed that the application of Gaussian noise (3–35 Hz) reduces tremor (measured with accelerometric amplitude and electromyography (EMG) activity) and improves the motor performance in individuals with enhanced physiological tremor [25]. It was also shown that Stochastic resonance of 0–15 Hz (a phenomenon in non-linear systems characterized by a response increase in the system induced by a particular level of input noise) improves the motor task in healthy individuals and that the higher degree SR is more pleasant with 0–300 Hz and 250–300 Hz noise bandwidths than for 0–15 Hz. The principle of action of the manipulandum used in these studies resembles that of the VILIM BALL [26]. Wrist tendon vibration (TV, 70 Hz) was applied to the forearm wrist musculature improved arm stability, as evidenced by the decreased magnitude of hand tangential velocity at the target. Improved stability was accompanied by a decrease in muscle activity throughout the arm, as well as a mean decrease in grip pressure in 10 hemiparetic stroke patients [27]. 

We did not identify major safety risks during the clinical study; however, the local hand arm vibration is associated with the development of certain conditions: Raynaud’s phenomenon, carpal tunnel syndrome [28], vibration-induced white finger disease [29], finger pain, back pain, muscular pain or fatigue [30], chronic subdural hematoma (local application to the head) [31], skin irritation (including redness, itchiness, and/or swelling) [24], finger blood flow reduction [30,31,32,33], and neck and upper limb musculoskeletal disorders. Vibration can increase the postural and rest tremor [34], exacerbate Dupuytren’s contracture, and have various osteoarticular effects (hand and carpal bone vacuoles and cysts, Kienbock’s disease, navicular pseudoarthrosis, olecranon spurs, and osteoarthrosis of the wrist and elbow joints). People with the known HTR1B gene may be more susceptible to the development of secondary Raynauds’ phenomenon due to vibration exposure [33,34,35,36]. Prolonged vibration may reduce the manual dexterity [37]. Systemic adverse effects that result from local hand-arm vibration may include a cardiovascular response (echocardiographic changes (an increase in ejection fraction and stroke volume, enlarged left ventricular diastolic dimension, and reduction in heart rate) and lower blood pressure) [38], immunological changes (e.g., an increase in T cell lymphocytes of CD4 and CD8), and suppression of serum total cholesterol and triglyceride levels [36,37,38]. Vibration may promote vascular injury, endothelial injury, microvasculature changes [39,40,41], defects in vascular repair, and intravascular abnormalities (release of vasoactive mediators) [42].

## 5. Conclusions

To conclude, we show that the investigated medical device reduces tremor in patients with essential tremor and that the device has a favorable benefit risk-ratio.

## Figures and Tables

**Figure 1 medicina-56-00552-f001:**
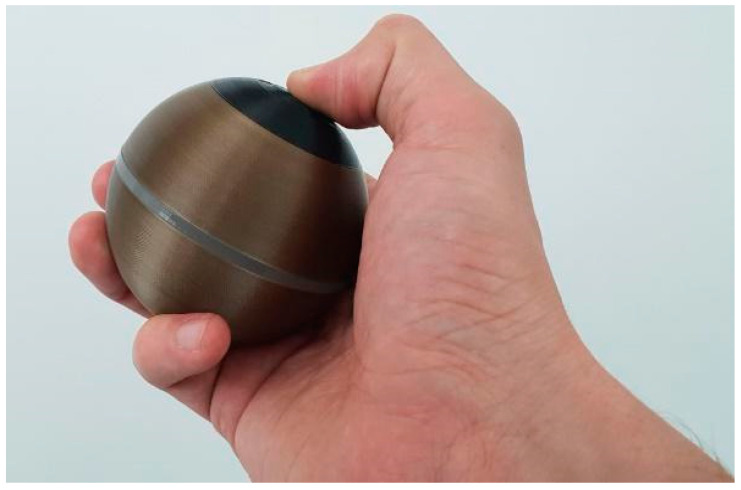
The Vilim ball device.

**Figure 2 medicina-56-00552-f002:**
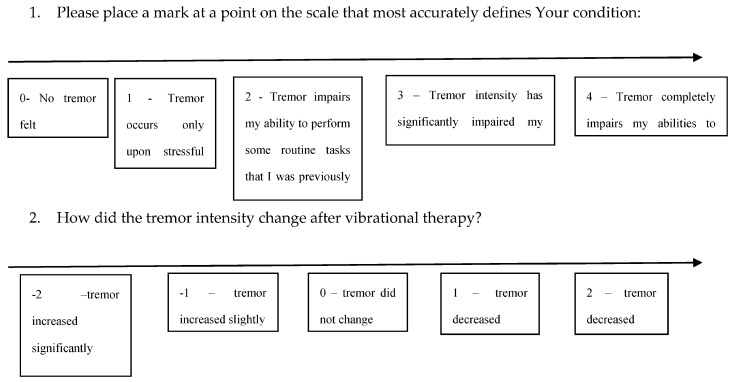
Secondary efficacy endpoint.

**Table 1 medicina-56-00552-t001:** Inclusion and exclusion criteria.

Inclusion Criteria
Male and female subjects 18–65 years of age at the time of consent; Written informed consent provided by the patient; Patients diagnosed with essential tremor.
**Exclusion Criteria**
Non-compliance; Substance-dependence; Uncontrolled medical illness (arterial hypertension, diabetes, severe liver or kidney failure); Subjects with following abnormal laboratory values at screening visit: Hemoglobin < 11.0 g/dL (<110.0 g/L) or hematocrit < 30% (<0.30 *v*/*v*), White blood cell count <3.0 × 109/L (<3000/mm^3^), Absolute neutrophil count of <1.5 × 109/L (<1500/mm^3^), Absolute lymphocyte count of <0.5 × 109/L (<500/mm^3^), Platelet count < 100 × 109/L, Estimated creatinine clearance < 40 mL/min based on the Cockcroft–Gault calculation or serum creatinine value greater than 1.5 times the upper limit of normal (ULN), Aspartate aminotransferase (AST) or alanine aminotransferase (ALT) values greater than 2 times the upper limit of normal, Subject has positive results for hepatitis B surface antigens (HBsAg), antibodies to hepatitis B core antigens (anti-HBc), hepatitis C virus (HCV), or human immunodeficiency virus (HIV); Other subjects who, in the opinion of the investigator, have any acute or chronic medical or psychiatric condition or laboratory abnormality at the screening that would impair the subject’s capability to follow the study protocol.

**Table 2 medicina-56-00552-t002:** IMD (investigated medical device) effectiveness and baseline tremor intensity as reported by patients.

	Tremor Increased Slightly	Tremor Did Not Change	Tremor Decreased Slightly
Tremor occurs only upon stressful situation	0	0	1
Tremor impairs my ability to perform some routine tasks that I was previously capable of performing	0	4	3
Tremor intensity has significantly impaired my ability to perform daily tasks	2	3	3

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
