# Peer review of "Local Vibrational Therapy for Essential Tremor Reduction: A Clinical Study"

_medicina, 2020, doi:10.3390/medicina56100552_

Round 1
Reviewer 1 Report
The article is disorganized and very difficult to follow. Data on the systematic review of the literature (not meta-analysis) on the usefullness of local vibrational therapy in several diseases should be removed from this article and the authors should focus on their effects of essential tremor.
There is a mixing between experimental data and clinical data through the methods and results section that should be avoided.
In the experimental data, the ethical statement should be mentioned after the description of the experimental procedure. In the description of the experimental procedure, the authors mention "...recording as previously described" without any reference to this previous work,
Minor comments: current and future options for the pharmacological treatment of essential tremor have been revised in a recent article (PMID 2777152). Recommendations of the Movement Disorders society regardint ET therapy (including surgery) (PMID 30146186)
Author Response
Question. The article is disorganized and very difficult to follow. Data on the systematic review of the literature (not meta-analysis) on the usefullness of local vibrational therapy in several diseases should be removed from this article and the authors should focus on their effects of essential tremor. There is a mixing between experimental data and clinical data through the methods and results section that should be avoided.
Answer. We agree with this preposition. We added the systematic review to support the need for a clinical trial. As our approach this seems to hinder understanding, the non-clinical data and systematic review will be only added as supplementary material because we do not seek to use that data for a separate publication. We added the systematic review to the Appendix B. We keep safety data of both conditions because we were reviewing mainly safety aspects of the local vibrational therapy. It is important to identify and describe safety aspects that were not noted during our study.
Question. In the experimental data, the ethical statement should be mentioned after the description of the experimental procedure. In the description of the experimental procedure, the authors mention "...recording as previously described" without any reference to this previous work,
Answer. Appropriate citation was added. The vascular reactivity study was added in the Appendix B and omitted from the main text.
Question. Minor comments: current and future options for the pharmacological treatment of essential tremor have been revised in a recent article (PMID 2777152). Recommendations of the Movement Disorders society regardint ET therapy (including surgery) (PMID 30146186)
Answer. We researched proposed sources, however PMID 2777152 and PMID 30146186 yielded articles that were rather out of scope:
However, as per Your comment I manged to find this article and enriched the introduction with it:
Alonso-Navarro H, García-Martín E, Agúndez JAG, Jiménez-Jiménez FJ. Current and Future Neuropharmacological Options for the Treatment of Essential Tremor. Curr Neuropharmacol. 2020;18(6):518-537. doi: 10.2174/1570159X18666200124145743. PMID: 31976837; PMCID: PMC7457404.

Reviewer 2 Report
I think this treatment is significant in that it may be temporary but very helpful if the patient has a resistance to the drug or if the drug does not work well.
- The clinical trials were conducted on Parkinson's patients, and how did this device affect Parkinson's patients with tremors ?
- Has there been any difference in symptom relief depending on stage of disease severity (e.g., if higher H-Y sage, is the effect weak ?)
- Is it possible to use it consecutively? (What is the cycle of use, and whether it doesn't change depending on the person.)
- What happens to patients with increased tremor and how long it lasts (whether this is a side effect)
Author Response
Question. I think this treatment is significant in that it may be temporary but very helpful if the patient has a resistance to the drug or if the drug does not work well.
Answer. We agree. This device can be used with drugs to increase their efficacy.
Question. The clinical trials were conducted on Parkinson's patients, and how did this device affect Parkinson's patients with tremors?
Answer. The main efficacy study did not include patients with parkinsonian tremor and we are not seeking this therapeutic indication. In the extended safety study 1 patient with parkinsonian tremor reported lack of efficacy in terms of symptom reduction and 1 – in terms of function improvement. The rest of the parkinsonian tremor patients reported either improvement in function, reduction is symptom severity or both.
Question. Has there been any difference in symptom relief depending on stage of disease severity (e.g., if higher H-Y sage, is the effect weak?)
Answer. We did not find that symptom severity would have any influence on the device’s efficacy.
Question. Is it possible to use it consecutively? (What is the cycle of use, and whether it doesn't change depending on the person.)
Answer. The hand-held device can be used as needed, up to 1 hour per day in total. Can be used on both hands as needed to decrease hand tremor. The device can be reused by the same user. Two devices can be used at the same time.
Question. What happens to patients with increased tremor and how long it lasts (whether this is a side effect)
Answer. The device may sometimes intensify tremor, but this is short lasting. This effect does not recur, no long term effects were observed and no patients reported this as a significant adverse effect.

Round 2
Reviewer 1 Report
No additional comments